# Leveraging Vision-Language Models for Resource Constrained Settings

**Anna Bair**                                                              *abair@cmu.edu*
*Machine Learning Department*
*Carnegie Mellon University*

**Madan Ravi Ganesh**                                    *Madan.RaviGanesh@us.bosch.com*
*Bosch Center for Artificial Intelligence*

**Devin Willmott**                                         *Devin.Willmott@us.bosch.com*
*Bosch Center for Artificial Intelligence*

**J. Zico Kolter**                                                      *zkolter@cs.cmu.edu*
*Machine Learning Department*
*Carnegie Mellon University*

**Reviewed on OpenReview:** *https://openreview.net/forum?id=cYOKSg6OjC*

## Abstract

Vision-language models (VLMs) such as CLIP have emerged as extremely strong zero-shot and few-shot image classifiers. However, these models are often too expensive or cumbersome for resource constrained downstream applications. In this work, we examine how to best leverage the strength of pretrained VLMs: by extracting *task-specific* information in order to obtain a small model that can be deployed in a very specific and low-resource setting. We present the SIDCLIP method, a training pipeline which drastically improves the performance of small, efficient models, such as EfficientNet B0. The pipeline includes three components that are critical to obtaining strong performance: 1) augmenting the classifier with *synthetic data* generated by leveraging CLIP itself; 2) *initializing* the modeling process using a smaller CLIP model pretrained on the target architecture; and 3) incorporating *knowledge distillation* to maximally mimic the performance of the larger model. SIDCLIP improves the performance of an EfficientNet B0 model by an average of 55 percentage points on 1-shot versions of four datasets and by an average of 29 points on the 8-shot versions, relative to directly trained networks, additionally approaching CLIP's linear probe performance while using a model with less than 2% of the parameters of CLIP ViT-L/14's image encoder. Our work is intended to be a practical guide for leveraging the power of foundation models in downstream data-scarce and budget constrained settings. Code is released at https://github.com/annaebair/sidclip.

## 1 Introduction

Foundation models such as CLIP-based models have been shown to perform extremely well on zero-shot and few-shot image classification: via simple prompting and/or a few examples, these models can achieve classification performance on par with models trained with much more task-specific data (Radford et al., 2021). However, this performance comes at a cost: the models are extremely general and large-scale, and thus incur a high inference cost relative to smaller, more task-specific models, making them unsuitable for many edge applications. This challenge has led to a number of methods for compressing or distilling knowledge from large foundation models into smaller models. Although these techniques can preserve strong performance relative to the large foundation model, they are often not task-specific, and when they are, they

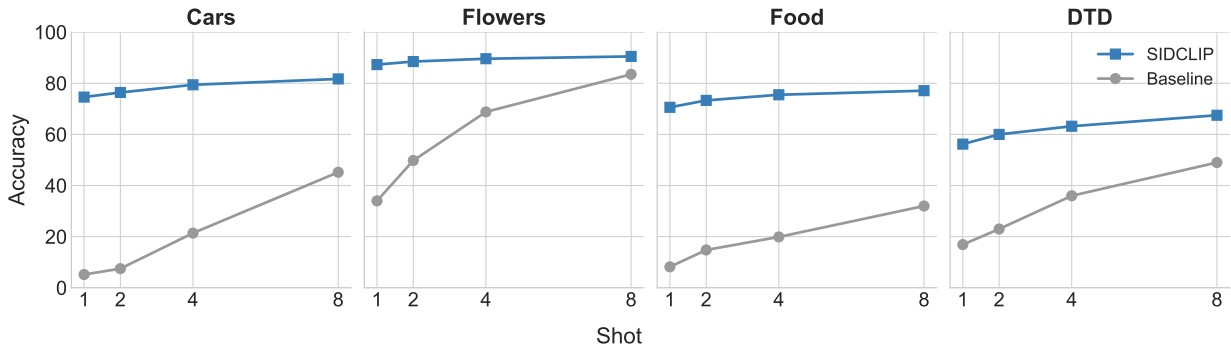

Figure 1: Using SIDCLIP to train an EfficientNet B0 model drastically outperforms standard training. Each plot includes results from a different dataset. "Baseline" refers to an EfficientNet B0 finetuned on few-shot data under a standard training regimen. "SIDCLIP" demonstrates the notable improvement when training with our proposed method.

often focus on preserving the model's zero-shot performance for new tasks, rather than taking advantage of limited task-specific downstream data (Popp et al., 2024; Li et al., 2023; Wu et al., 2023; Vasu et al., 2024; Sun et al., 2023).

In this work, our goal is to produce a small model that performs as close as possible to a powerful large-scale vision-language model (VLM) on a particular downstream task. We address the specific challenge of attempting to leverage the strong performance of zero- and few-shot CLIP image classification models into vastly more efficient (but task-specific) architectures. In other words, given a very limited amount of data on a desired downstream image classification task, and a very limited inference-time compute budget, we obtain the best performance on a downstream compact model by *leveraging the capabilities of larger models*. In practice, we find that three separate components are central to obtaining strong performance:

1. We augment the classifier with **synthetic data** generated by leveraging CLIP itself. Specifically, we use a text-to-image generative model seeded with embeddings produced from linear interpolations of the text of the class label *and* the CLIP image embeddings of the available data.

2. We **initialize** our small model as a variant based on a small CLIP model pretrained on the target architecture.

3. We incorporate **knowledge distillation** to maximally mimic the performance of the larger CLIP model.

We call our method, which incorporates the above three components, *SIDCLIP* (Synthesize-Initialize-Distill CLIP). While each of these elements alone have been the subject of exploration in the literature, we emphasize that the work here serves largely as a practical guide that demonstrates the relative value of leveraging these three capabilities, as well as ablations demonstrating the efficacy of subsets of these elements.

We evaluate our proposed approach, along with ablations and other baselines, on four common small-scale image classification benchmarks: the Stanford Cars (Krause et al., 2013), Oxford Flowers (Nilsback & Zisserman, 2008), Food 101 (Bossard et al., 2014), and Describable Textures (DTD) (Cimpoi et al., 2014) datasets. We improve the performance of small models by up to 69 percentage points across a range of models and datasets. SIDCLIP-trained models approach and even exceed the teacher's performance. A comparison of SIDCLIP training and standard training is shown in Figure 1. For a fixed student (EfficientNet B0), we show the difference in performance across datasets between standard training ("Baseline") and the best performing SIDCLIP variant.

## 2 Related Works

**Synthetic data.** There has been notable evidence to indicate that synthetic data is helpful in general when training models and particularly in distillation. Azizi et al. (2023) find that augmentation of a dataset with synthetic data improves image classification performance on CNN and ViT architectures. He et al. (2023) focus on the zero- and few-shot domains and reaches a similar conclusion: that synthetic data can be used in conjunction with real data to improve performance on image classification tasks. Similarly to our work, Popp et al. (2024) generate synthetic data in order to perform distillation. This work differs from ours in two notable ways: they assume *no* access to the downstream data, rather than a small number of samples; and the aim is to transfer the general zero-shot capabilities of CLIP rather than focusing on a particular downstream task. More generally, this area of data-free distillation explores the usage of only synthetic data (and no real data) during the distillation process (Chawla et al., 2021; Fang et al., 2022).

While these prior works all incorporate synthetic data, none utilize the particular image- and text-conditioning generation method that we use in SIDCLIP. The image generation pipeline we use was introduced in Razzhigaev et al. (2023) and achieves SOTA FID scores on generated images relative to other open source models.

**Compression.** VLMs have remarkable few- and zero-shot performance on downstream tasks and are strong image classifiers (Radford et al., 2021; Jia et al., 2021; Li et al., 2022; Yuan et al., 2021; Zhai et al., 2023). There has been a range of work exploring the natural next step of attempting to compress these high powered models into smaller versions that require less memory and have lower inference times. Some (such as pruning, quantization, and distillation) mirror compression in non-foundation models, while others (including parameter-efficient fine-tuning such as adapter layers or prompt tuning) are unique to the VLM or LLM setting (Hinton et al., 2015; Dettmers et al., 2022; Frantar & Alistarh, 2023; Sun et al., 2024; Houlsby et al., 2019; Liu et al., 2022; Lester et al., 2021; Jia et al., 2022).

In many of the existing efforts to compress foundation models, the goal has been to preserve the *general* capabilities of the models. Rather than focusing on a model's performance on a particular task, these methods aim to broadly preserve the VLM's generalization abilities for image classification (Li et al., 2023; Wu et al., 2023; Vasu et al., 2024; Sun et al., 2023; Wu et al., 2022; Cai et al., 2025).

TinyCLIP and MobileCLIP both preserve CLIP's general purpose knowledge through distillation (Wu et al., 2023; Vasu et al., 2024). TinyViT is another method which produces a small downstream model via distillation (Wu et al., 2022). Task-specificity is not part of the distillation process for any of these methods.

Similarly to our CLIP-initialized small model, Sun et al. (2023), distill from CLIP ViT-L/14 to a smaller foundation model, and find that this distilled model outperforms a similar model trained from scratch. However, their smallest model (Swin-T) is over three times larger than our largest model and they examine only the zero-shot setting. The value of knowledge distillation for task-specific small model performance is also highlighted in Jang et al. (2025), but they do not explore few-shot settings.

Li et al. (2023) distills from a CLIP ViT-L/14 teacher to a convolutional network student such as ResNet18. They measure task-specific performance as out-of-distribution performance: they perform distillation without any of the task-specific samples and then evaluate the zero- or few-shot performance of their model on downstream tasks. While similar to our setting, this setting does not take advantage of task-specific data available during distillation and thus yields lower performance than our method.

**Few-shot learning.** While preserving the entirety of CLIP's performance is a worthwhile goal, it is not the correct focus for all settings. The few-shot setting, when there is limited downstream training data available, arises in situations where data collection is expensive or challenging (Wang et al., 2020). Training large-scale models from scratch is an extremely data-intensive process, so usage of few-shot data to finetune an existing model can increase accessibility and customization of the power of VLMs. While there is some work that addresses a few-shot downstream setting, these methods often preserve or augment the network architecture of large CLIP models, thus making these approaches less feasible solutions for resource-constrained users

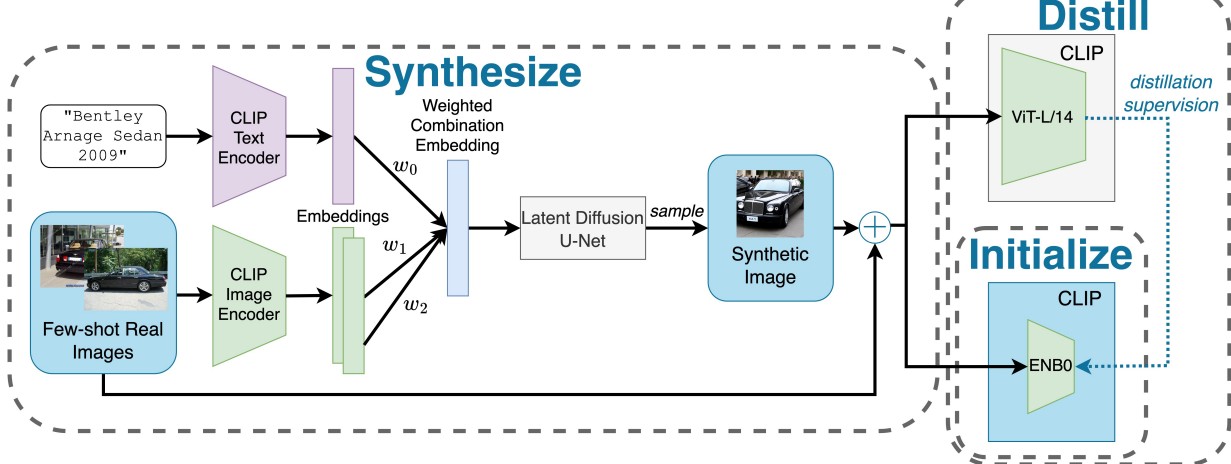

Figure 2: The three components of SIDCLIP: *synthesize* data via a weighted combination of class labels and real images; *initialize* the student as the image encoder of a small CLIP model; *distill* from a powerful teacher model.

(Ma et al., 2024; Wortsman et al., 2022; Islam et al., 2021). If some downstream task-specific training data is available, these methods are not equipped to best utilize it.

## 3 The SIDCLIP Method

**Motivation.** To use CLIP as an image classifier, first an image is passed into the image encoder, and text of the possible classnames is passed into the text encoder. Then, the embedding similarity between the image and each possible classname is measured. Although this process yields high accuracy on a variety of downstream tasks, the CLIP model is unnecessarily large for many downstream applications, such as use on edge devices: one of the most commonly used CLIP models, CLIP ViT-L/14, has 307M parameters in its image encoder (Radford et al., 2021).

Here we ask: what if a user wants to take advantage of CLIP's strong off-the-shelf zero and few shot performance but does not need its full "general-purpose" abilities? They may only need to classify images corresponding to a specific task and cannot afford to run such a large model. In this case, it is desirable to transfer only a specific portion of CLIP's image classification capabilities to a smaller model.

With existing methods such as TinyCLIP and MobileCLIP, a user would be able to produce a general-purpose small model, and potentially finetune it on the task of interest, but is left without being able to optimally take advantage of the limited training data they have (Wu et al., 2023; Vasu et al., 2024). They would end up with a smaller version of CLIP, not a model tailored to their specific use case.

**Problem setting.** Our goal is to maximize the performance on a particular image classification task, subject to resource constraints. We have access to a small model $\mathcal{S}$ that fits our budget constraints and $k$ labeled samples per class $c \in \mathcal{C}$, for $n = |\mathcal{C}|$ classes. We also have access to a large scale teacher VLM $\mathcal{T}$, such as a CLIP model.

Our method, SIDCLIP, consists of three essential components for leveraging CLIP's power in training a small model in a data-constrained setting. These three components are 1) synthetic data, 2) initializing the model as a small CLIP variant, and 3) distilling from CLIP to the small model. The pipeline is shown in Figure 2. We present a practical guide to training small models in resource-constrained settings, with code available at https://github.com/annaebair/sidclip.

### 3.1 Component #1: Synthetic Data

We use synthetic data to augment the limited samples per class in a few shot setting. As described in our problem setting, we have $k$ labeled samples per class. We use these $k \times n$ samples $\mathcal{D}_r$ and their classname labels $\mathcal{L}$ to generate additional synthetic samples that can be used for training the model. When operating in the $k$-shot setting, we *only* use those $k$ samples and their classnames as input to generate additional synthetic data. The generative diffusion model we use accepts CLIP text and/or image embeddings as input and conditions its generation on these inputs. This generative process allows us to extract task-specific data from the teacher CLIP model.

More formally, when we want to generate a synthetic sample from class $c$, we use the label $l_c \in \mathcal{L}$ and some set of images $\{x_i\}_{i=1}^I \in \mathcal{D}_{r,c}$, for $I \leq k$ and where $\mathcal{D}_{r,c}$ refers to the real data available from class $c$. We obtain the CLIP image and text embeddings: $\texttt{img\_emb}(x_i)$ and $\texttt{text\_emb}(l_c)$ and combine them via a weighted combination:

$$\texttt{emb} = w_0 \cdot \texttt{text\_emb}(l_c) + \sum_{i=1}^I w_i \cdot \texttt{img\_emb}(x_i) \tag{1}$$

such that $\sum_{i=0}^I w_i = 1$. This combination is then passed into the generative model, which we sample from to obtain $J$ synthetic samples $\mathcal{D}_{s(r)} = \{x'_j\}_{j=1}^J \sim \mathcal{G}(\texttt{emb})$. The subscript $s(r)$ emphasizes that the synthetic samples are generated using only the real images $\mathcal{D}_r$ and their labels.

In most cases where synthetic data is used for training, images are generated based on solely a text prompt or a text prompt and an existing image (see Section 2). In this work, we aim to maximally leverage the existing data by utilizing a data generation pipeline which can take as input linear combinations of embeddings of text and *multiple* images.

Concretely, we use the Kandinsky framework, which takes as input real images and captions (Razzhigaev et al., 2023). This pipeline obtains CLIP embeddings for each image and caption, combines them according to specified weights, and passes the joint embedding into the diffusion model to produce a synthetic sample. We chose this pipeline due to its high performance, flexibility, and off-the-shelf ease of use: it achieved strong FID scores relative to competitors and was the first text-to-image generative model that used both image priors and latent diffusion.

### 3.2 Component #2: Initialize as Small CLIP

We find that initializing a student model in a CLIP-style architecture allows for performance gains relative to a standalone student vision model. In this paper, we distill to three models in the EfficientNet family, a set of small convolutional networks (Tan & Le, 2020). Specifically, we use EfficientNet B0, B1, and B2, with parameter counts of 5.3M, 7.8M, and 9.2M, respectively. For our primary set of experiments, we initialize these models as small CLIP variants, that is, preserve the CLIP text encoder and replace the CLIP image encoder with the EfficientNet model. Each CLIP-EfficientNet model is pretrained on a subset of the DataComp dataset (Gadre et al., 2023).

### 3.3 Component #3: Knowledge Distillation

Knowledge distillation is a common model compression technique that uses a large, powerful teacher model to train a smaller student model by aligning the student's output probabilities to those of the teacher. There are many variants of loss functions used to align these sets of probabilities, but the most common is based on the Kullback-Leibler (KL) divergence as proposed in Hinton et al. (2015):

$$\mathcal{L}_{KL} = \alpha \cdot T^2 \cdot D_{\text{KL}}(SM(\tilde{y}/T), SM(\hat{y}/T)) + (1 - \alpha) \cdot CE(\hat{y}, y) \tag{2}$$

where $D_{\text{KL}}$ refers to KL divergence, $CE$ is cross entropy, $SM$ is softmax, $\tilde{y}$ is the teacher output probabilities, $\hat{y}$ is the student output probabilities, $y$ is the true labels, $\alpha$ is a hyperparameter that trades off influence from teacher labels and true labels, and $T$ is a temperature parameter.

We use this standard KL setting in our experiments. We have a teacher image encoder which outputs image embeddings of size $d_{img}^T$ and a student image encoder which outputs image embeddings of size $d_{img}^S$. We also have a common text encoder which produces text embeddings of size $d_{text}$.

For each task, we append classification heads to both teacher and student models, with linear layers of shape $d_{img}^T \times c$ and $d_{img}^S \times c$, respectively. Before distillation, we finetune the teacher linear layer on the task of interest. We initialize the student layer with the text embeddings of each class: we obtain the embeddings for captions `"A photo of {classname}."` or `"A photo of {classname}, a type of {category}."` for each class and concatenate them into a tensor of shape $d_{text} \times c$ where $d_{img}^S = d_{text}$. Then, during distillation, the teacher and its appended layer are frozen, and both the student and its appended layer are updated.

We use a distillation set $\mathcal{D} = \mathcal{D}_r \cup \mathcal{D}_{s(r)}$ consisting of the real samples $\mathcal{D}_r$ and the synthetic samples $\mathcal{D}_{s(r)}$ generated by conditioning on those real images.

# 4 Results

We demonstrate that SIDCLIP allows us to approach the performance of CLIP ViT models while using an image encoder with as few as 2% of the parameters. Each of the three components (synthesize, initialize, distill) is critical in achieving this strong performance. Through a series of ablations and comparisons to SOTA distillation methods, we show that SIDCLIP is the dominant method when operating in a resource constrained setting.

## 4.1 Experimental Details

Here we include the main settings for our experiments. Additional hyperparameter details can be found in Appendix A.3.

**Datasets.** We report results on four task-specific image classification datasets: StanfordCars, OxfordFlowers, Food101, and DTD (Krause et al., 2013; Nilsback & Zisserman, 2008; Bossard et al., 2014; Cimpoi et al., 2014). We chose these datasets due to the fine-grained nature of their classification tasks. Unlike more general classification datasets such as ImageNet, these datasets are restricted to a very limited domain, and are similar to very specific classification tasks that an end user may want to perform. StanfordCars has 196 classes, OxfordFlowers has 102, Food101 has 101, and DTD has 47. All numbers reported in this paper are Top 1 accuracies on the test sets.

**Data-scarce setting.** We are generally interested in any limited data setting. For experimental purposes, we simulate a data-scarce setting by creating few shot datasets from existing task-specific datasets. For each dataset, we randomly sample $k$ images per class to produce a $k$-shot variant of the dataset, for $k = \{1, 2, 4, 8\}$.

**Synthetic data.** We generate 300 synthetic samples per class for all few shot settings, and 100 samples for the zero-shot setting. In all of the few-shot settings, our distillation dataset includes the few real samples per class and the synthetic samples generated from only those real samples. See Section A.2 for more details.

**Models.** We use two CLIP models as teachers, CLIP-ViT-L/14 and CLIP ViT-B/32 (Radford et al., 2021). We also use three EfficientNet models (B0, B1, and B2) as students, each initialized in a CLIP-style model (Tan & Le, 2020; Akinwande et al., 2024). When performing distillation, the teacher is frozen and we update the parameters of both the student model's image encoder and its appended linear layer. When performing finetuning of a CLIP-style model we freeze the parameters of the student model and only update the parameters in the appended linear layer. In particular, when we finetune the teacher CLIP models, the linear classification layer is finetuned once per dataset on the full training set, not on the few-shot subset, and is reused across all shot counts and student models. When finetuning or distilling to a non-CLIP-style model, there is no appended linear layer and we update all model parameters.

Due to computational constraints, the CLIP-EfficientNet B1 and CLIP-EfficientNet B2 models were pre-trained with less DataComp data than the CLIP-EfficientNet B0 model. Therefore, we note that while results within each EfficientNet model consistently demonstrate our findings, results *across* EfficientNet

models are not necessarily comparable. In order to provide some comparison, we include correlations of results across models in Table 6.

Table 1: SIDCLIP outperforms baselines and competing methods. Few-shot train and SIDCLIP results are mean $\pm$ std over 3 seeds.

| Dataset | Model | Method | Params (M) | Shot | | | | |
| | | | | 1 | 2 | 4 | 8 | Full |
|---|---|---|---|---|---|---|---|---|
| Cars | CLIP ViT-B/32 | FT | 86 | 43.9 | 48.5 | 58.6 | 65.9 | 80.2 |
| | CLIP ViT-L/14 | FT | 307 | 78.1 | 79.0 | 81.5 | 83.3 | 91.1 |
| | | Train | 5.3 | $5.1 \pm 0.2$ | $7.9 \pm 0.3$ | $19.7 \pm 0.5$ | $43.6 \pm 1.2$ | **87.6** |
| | EfficientNet B0 | SIDCLIP$_{B/32}$ | 5.3 | $\mathbf{73.7} \pm 0.6$ | $\mathbf{75.5} \pm 0.5$ | $\mathbf{79.1} \pm 0.4$ | $\mathbf{81.6} \pm 0.2$ | 85.5 |
| | | SIDCLIP$_{L/14}$ | 5.3 | $66.1 \pm 1.4$ | $69.3 \pm 2.1$ | $76.3 \pm 1.4$ | $81.4 \pm 0.6$ | 87.3 |
| | | Train | 7.8 | $4.7 \pm 0.2$ | $8.1 \pm 0.4$ | $17.9 \pm 0.1$ | $41.5 \pm 0.7$ | **87.6** |
| | EfficientNet B1 | SIDCLIP$_{B/32}$ | 7.8 | $\mathbf{71.4} \pm 0.6$ | $\mathbf{72.8} \pm 0.5$ | $\mathbf{76.9} \pm 0.3$ | $\mathbf{79.8} \pm 0.3$ | 84.0 |
| | | SIDCLIP$_{L/14}$ | 7.8 | $59.9 \pm 1.8$ | $63.5 \pm 2.3$ | $72.2 \pm 1.1$ | $78.5 \pm 0.6$ | 85.3 |
| | | Train | 9.2 | $4.6 \pm 0.3$ | $7.0 \pm 0.4$ | $18.7 \pm 0.4$ | $44.6 \pm 0.4$ | **88.3** |
| | EfficientNet B2 | SIDCLIP$_{B/32}$ | 9.2 | $\mathbf{71.9} \pm 0.9$ | $\mathbf{73.4} \pm 0.5$ | $\mathbf{77.5} \pm 0.5$ | $\mathbf{79.9} \pm 0.2$ | 84.0 |
| | | SIDCLIP$_{L/14}$ | 9.2 | $61.0 \pm 1.7$ | $64.3 \pm 1.7$ | $73.7 \pm 1.1$ | $79.1 \pm 0.5$ | 85.2 |
| Flowers | CLIP ViT-B/32 | FT | 86 | 65.0 | 76.6 | 86.4 | 90.8 | 92.8 |
| | CLIP ViT-L/14 | FT | 307 | 90.3 | 94.9 | 97.5 | 98.5 | 98.6 |
| | | Train | 5.3 | $34.4 \pm 1.6$ | $53.1 \pm 2.6$ | $68.0 \pm 0.7$ | $83.2 \pm 0.9$ | 86.9 |
| | EfficientNet B0 | SIDCLIP$_{B/32}$ | 5.3 | $\mathbf{88.0} \pm 0.5$ | $88.7 \pm 0.1$ | $90.0 \pm 0.8$ | $91.0 \pm 0.2$ | 91.4 |
| | | SIDCLIP$_{L/14}$ | 5.3 | $87.6 \pm 0.5$ | $\mathbf{89.9} \pm 0.4$ | $\mathbf{93.3} \pm 0.4$ | $\mathbf{94.9} \pm 0.4$ | **95.7** |
| | | Train | 7.8 | $35.9 \pm 0.7$ | $54.1 \pm 1.7$ | $70.8 \pm 1.1$ | $84.1 \pm 1.2$ | 88.1 |
| | EfficientNet B1 | SIDCLIP$_{B/32}$ | 7.8 | $\mathbf{85.8} \pm 0.3$ | $86.0 \pm 1.2$ | $88.2 \pm 0.2$ | $90.0 \pm 0.2$ | 89.7 |
| | | SIDCLIP$_{L/14}$ | 7.8 | $84.5 \pm 0.9$ | $\mathbf{86.7} \pm 0.2$ | $\mathbf{90.9} \pm 0.2$ | $\mathbf{93.4} \pm 0.1$ | **93.1** |
| | | Train | 9.2 | $32.5 \pm 2.2$ | $51.3 \pm 1.0$ | $66.6 \pm 1.8$ | $84.4 \pm 0.5$ | 87.7 |
| | EfficientNet B2 | SIDCLIP$_{B/32}$ | 9.2 | $\mathbf{86.1} \pm 0.4$ | $86.8 \pm 0.9$ | $88.6 \pm 0.7$ | $90.6 \pm 0.0$ | 89.7 |
| | | SIDCLIP$_{L/14}$ | 9.2 | $84.7 \pm 0.3$ | $\mathbf{87.9} \pm 0.5$ | $\mathbf{91.4} \pm 0.6$ | $\mathbf{94.1} \pm 0.1$ | **94.1** |
| Food | CLIP ViT-B/32 | FT | 86 | 66.5 | 70.7 | 74.5 | 78.3 | 86.7 |
| | CLIP ViT-L/14 | FT | 307 | 92.8 | 92.8 | 93.1 | 93.4 | 95.2 |
| | | Train | 5.3 | $8.2 \pm 0.8$ | $13.3 \pm 0.6$ | $19.1 \pm 0.7$ | $30.0 \pm 0.8$ | 83.3 |
| | EfficientNet B0 | SIDCLIP$_{B/32}$ | 5.3 | $\mathbf{70.0} \pm 0.5$ | $\mathbf{72.4} \pm 0.1$ | $\mathbf{75.1} \pm 0.3$ | $\mathbf{77.0} \pm 0.0$ | 86.6 |
| | | SIDCLIP$_{L/14}$ | 5.3 | $58.7 \pm 1.5$ | $64.0 \pm 1.2$ | $69.3 \pm 0.7$ | $72.8 \pm 0.7$ | **88.9** |
| | | Train | 7.8 | $9.3 \pm 0.5$ | $14.4 \pm 0.9$ | $20.6 \pm 0.1$ | $31.7 \pm 0.3$ | 83.9 |
| | EfficientNet B1 | SIDCLIP$_{B/32}$ | 7.8 | $\mathbf{65.9} \pm 0.3$ | $\mathbf{68.4} \pm 0.5$ | $\mathbf{71.2} \pm 0.4$ | $\mathbf{73.7} \pm 0.1$ | 85.6 |
| | | SIDCLIP$_{L/14}$ | 7.8 | $52.6 \pm 1.6$ | $57.9 \pm 1.1$ | $64.6 \pm 1.0$ | $68.4 \pm 0.7$ | **87.9** |
| | | Train | 9.2 | $7.6 \pm 0.7$ | $12.5 \pm 0.5$ | $18.6 \pm 0.2$ | $29.7 \pm 0.2$ | 83.7 |
| | EfficientNet B2 | SIDCLIP$_{B/32}$ | 9.2 | $\mathbf{66.8} \pm 0.1$ | $\mathbf{68.8} \pm 0.4$ | $\mathbf{71.6} \pm 0.5$ | $\mathbf{74.0} \pm 0.2$ | 85.9 |
| | | SIDCLIP$_{L/14}$ | 9.2 | $53.7 \pm 1.5$ | $58.4 \pm 1.1$ | $64.9 \pm 1.0$ | $68.9 \pm 0.7$ | **88.2** |
| DTD | CLIP ViT-B/32 | FT | 86 | 47.0 | 54.3 | 58.6 | 64.4 | 73.0 |
| | CLIP ViT-L/14 | FT | 307 | 56.7 | 61.7 | 67.9 | 72.5 | 79.4 |
| | | Train | 5.3 | $19.0 \pm 0.6$ | $26.6 \pm 1.5$ | $34.1 \pm 0.6$ | $43.2 \pm 1.2$ | 63.5 |
| | EfficientNet B0 | SIDCLIP$_{B/32}$ | 5.3 | $\mathbf{56.6} \pm 0.6$ | $\mathbf{59.2} \pm 1.4$ | $\mathbf{63.6} \pm 1.2$ | $\mathbf{67.5} \pm 0.9$ | **71.7** |
| | | SIDCLIP$_{L/14}$ | 5.3 | $52.4 \pm 2.6$ | $55.4 \pm 2.1$ | $61.0 \pm 2.3$ | $66.8 \pm 0.9$ | **71.7** |
| | | Train | 7.8 | $20.3 \pm 1.2$ | $28.4 \pm 1.2$ | $34.4 \pm 0.1$ | $45.5 \pm 0.6$ | 63.8 |
| | EfficientNet B1 | SIDCLIP$_{B/32}$ | 7.8 | $\mathbf{56.6} \pm 1.4$ | $\mathbf{58.8} \pm 1.1$ | $\mathbf{62.8} \pm 1.7$ | $\mathbf{65.9} \pm 0.9$ | 68.7 |
| | | SIDCLIP$_{L/14}$ | 7.8 | $52.0 \pm 1.5$ | $55.9 \pm 3.0$ | $60.6 \pm 2.4$ | $64.8 \pm 0.6$ | **70.1** |
| | | Train | 9.2 | $18.6 \pm 0.3$ | $28.4 \pm 1.1$ | $34.4 \pm 0.6$ | $44.9 \pm 0.8$ | 65.2 |
| | EfficientNet B2 | SIDCLIP$_{B/32}$ | 9.2 | $\mathbf{57.8} \pm 1.5$ | $\mathbf{59.4} \pm 0.7$ | $\mathbf{63.6} \pm 1.1$ | $\mathbf{66.1} \pm 1.2$ | 68.9 |
| | | SIDCLIP$_{L/14}$ | 9.2 | $52.4 \pm 2.5$ | $56.6 \pm 1.9$ | $61.6 \pm 1.4$ | $65.9 \pm 1.0$ | **70.3** |

**Data augmentation.** We use RandAugment data augmentation (Cubuk et al., 2019). This is an augmentation strategy that applies random data augmentations to each image and is a top performing augmentation strategy. We apply six augmentations per image.

**Zero-shot results.** The zero-shot columns in Table 2 and Table 4 always indicates that no real data was used. For our method (the SIDCLIP rows), zero-shot distillation is performed by using 100 synthetic samples generated from only caption information. Lack of zero-shot results due to model incompatibility, unreleased results, or lack of synthetic data, is indicated by a dash ($-$).

## 4.2   Main Results

Across several teacher models, small student models, and few-shot dataset variants, SIDCLIP-trained models significantly outperform models trained via standard finetuning. Table 1 compares the results of applying SIDCLIP to those of training small models from scratch. For each dataset, we include upper bound baselines of CLIP ViT-L/14 and CLIP ViT-B/32 finetuned on the few-shot datasets. For each student architecture (EfficientNet B0, EfficientNet B1, and EfficientNet B2), we include one row indicating the results of standard finetuning ("Train") and one row with the results of training using SIDCLIP with each teacher (SIDCLIP$_{B/32}$ for CLIP ViT-B/32 and SIDCLIP$_{L/14}$ for CLIP ViT-L/14). Results are reported as average $\pm$ standard deviation across three random seeds.

Our goal was to leverage the power of CLIP to produce a strong small-scale model, using only limited training data. Our results indicate that, using each of our three components (synthesize, initialize, distill), we are able to obtain notable performance increases of up to 69 percentage points higher than the starting models in the few shot setting, with performances that approach and even exceed those of the teacher CLIP models. These findings generally hold across teacher and student models, datasets, and few-shot instances.

On the Cars, Flowers, and DTD datasets, SIDCLIP consistently achieves within around 10-30% of its teacher's performance in the few shot setting. On Food, SIDCLIP remains farther from the teacher model, particularly when distilling from CLIP ViT-L/14. We hypothesize that this may be due to more instances of food in the pretraining datasets for both teacher and student. In this case, additional food examples do not add much information to the model.

SIDCLIP with distillation from CLIP ViT-B/32 often outperforms distillation from CLIP ViT-L/14. We hypothesize this is due to the teacher-student capacity gap: the gap between a 307M parameter teacher and a 5.3M parameter student is much larger than between an 86M teacher and the same student, and prior work on distillation has shown that larger capacity gaps make knowledge transfer more difficult. The student may struggle to absorb the richer representations of the larger teacher, while the smaller teacher provides supervision that is closer to what the student can express.

## 4.3   Additional Comparisons

In Table 2 we include comparisons to other similar methods. Rows that include the performance of our proposed method are highlighted. TinyCLIP and TinyViT use distillation to train a downstream image classification model (Wu et al., 2022; 2023). Unlike our method, which allows for specialization on a specific task, these methods focus on maintaining CLIP's overall performance. Since few-shot results were not reported in these papers, we perform an evaluation of some of these methods. For TinyCLIP, we ran few-shot linear probe experiments on the smallest available model (8M parameter image encoder). For TinyViT, we ran few-shot finetuning experiments on the smallest available model (5.4M parameters).

Our method outperforms competitors by large margins in the few-shot setting. Although SIDCLIP performs worse than competitors on zero-shot, we note that the other models here are up to two times larger, and our strong few-shot results highlight the value of our pipeline in the intended setting.

We additionally include a limited comparison to parameter efficient finetuning methods Tip-Adapter and Tip-Adapter-F applied to our CLIP-EfficientNet B0 model in Table 3 (Zhang et al., 2022). We report mean and standard deviation across three random seeds. SIDCLIP generally outperforms both Tip Adapter variants across datasets and number of shots.

Table 2: SIDCLIP outperforms similar methods in the few shot setting. Results on one random seed.

| Model | Params (M) | Zero shot | Few shot (k) | Full shot |
|---|---|---|---|---|
| **Cars** | | | | |
| EfficientNet B0 (SIDCLIP) | 5.3 | 65.3 | **79.4** (4) | 85.5 |
| TinyViT-5M Wu et al. (2022) | 5.4 | - | 13.9 (4) | 87.7 |
| EfficientNet B1 (SIDCLIP) | 7.8 | 60.4 | 76.9 (4) | 84.0 |
| TinyCLIP Wu et al. (2023) | 8 | 7.8 | 17.1 (4) | 31.1 |
| EfficientNet B2 (SIDCLIP) | 9.2 | 61.6 | 72.3 (4) | 85.2 |
| TinyViT Popp et al. (2024) | 11 | **81.9** | − | **90.7** |
| ResNet18 Li et al. (2023) | 11 | 20.4 | 39.7 (5) | − |
| **Flowers** | | | | |
| EfficientNet B0 (SIDCLIP) | 5.3 | 10.1 | **92.6** (4) | **95.7** |
| TinyViT-5M Wu et al. (2022) | 5.4 | - | 74.9 (4) | 92.3 |
| EfficientNet B1 (SIDCLIP) | 7.8 | 4.8 | 90.0 (4) | 93.1 |
| TinyCLIP Wu et al. (2023) | 8 | 56.5 | 86.8 (4) | 82.4 |
| EfficientNet B2 (SIDCLIP) | 9.2 | 6.4 | 91.3 (4) | 94.1 |
| TinyViT Popp et al. (2024) | 11 | **68.3** | − | 90.6 |
| ResNet18 Li et al. (2023) | 11 | 18.2 | 54.3 (5) | − |
| **Food** | | | | |
| EfficientNet B0 (SIDCLIP) | 5.3 | 61.9 | **75.5** (4) | **86.6** |
| TinyViT-5M Wu et al. (2022) | 5.4 | - | 21.0 (4) | 84.7 |
| EfficientNet B1 (SIDCLIP) | 7.8 | 55.7 | 72.2 (4) | 85.6 |
| TinyCLIP Wu et al. (2023) | 8 | 55.1 | 58.4 (4) | 72.7 |
| EfficientNet B2 (SIDCLIP) | 9.2 | 56.6 | 72.3 (4) | 86.0 |
| TinyViT Popp et al. (2024) | 11 | **71.9** | − | 83.0 |
| ResNet18 Li et al. (2023) | 11 | 35.7 | 44.0 (5) | − |

Table 3: SIDCLIP outperforms Tip-Adapter variants in the few-shot setting. All results are mean $\pm$ std over 3 seeds.

| Dataset | Method | Shot | | | |
| | | 1 | 2 | 4 | 8 |
|---|---|---|---|---|---|
| Cars | Tip | $64.4 \pm 0.5$ | $66.4 \pm 0.3$ | $69.0 \pm 0.5$ | $72.2 \pm 0.2$ |
| | Tip-F | $65.4 \pm 0.2$ | $68.0 \pm 0.4$ | $70.4 \pm 0.7$ | $73.8 \pm 0.5$ |
| | SIDCLIP$_{B/32}$ | $\mathbf{73.7} \pm 0.6$ | $\mathbf{75.5} \pm 0.5$ | $\mathbf{79.1} \pm 0.4$ | $\mathbf{81.6} \pm 0.2$ |
| | SIDCLIP$_{L/14}$ | $66.1 \pm 1.4$ | $69.3 \pm 2.1$ | $76.3 \pm 1.4$ | $81.4 \pm 0.6$ |
| Flowers | Tip | $75.2 \pm 1.1$ | $79.2 \pm 1.4$ | $83.7 \pm 0.6$ | $86.1 \pm 0.4$ |
| | Tip-F | $76.4 \pm 1.4$ | $80.2 \pm 0.4$ | $84.4 \pm 1.3$ | $91.0 \pm 0.2$ |
| | SIDCLIP$_{B/32}$ | $\mathbf{88.0} \pm 0.5$ | $88.7 \pm 0.1$ | $90.0 \pm 0.8$ | $91.0 \pm 0.2$ |
| | SIDCLIP$_{L/14}$ | $87.6 \pm 0.5$ | $\mathbf{89.9} \pm 0.4$ | $\mathbf{93.3} \pm 0.4$ | $\mathbf{94.9} \pm 0.4$ |
| Food | Tip | $65.1 \pm 0.1$ | $65.1 \pm 0.3$ | $65.5 \pm 0.1$ | $65.8 \pm 0.1$ |
| | Tip-F | $65.2 \pm 0.0$ | $65.5 \pm 0.4$ | $66.4 \pm 0.1$ | $67.0 \pm 0.1$ |
| | SIDCLIP$_{B/32}$ | $\mathbf{70.0} \pm 0.5$ | $\mathbf{72.4} \pm 0.1$ | $\mathbf{75.1} \pm 0.3$ | $\mathbf{77.0} \pm 0.0$ |
| | SIDCLIP$_{L/14}$ | $58.7 \pm 1.5$ | $64.0 \pm 1.2$ | $69.3 \pm 0.7$ | $72.8 \pm 0.7$ |
| DTD | Tip | $35.8 \pm 0.6$ | $40.5 \pm 0.7$ | $46.5 \pm 1.1$ | $53.2 \pm 0.9$ |
| | Tip-F | $36.7 \pm 0.9$ | $45.3 \pm 0.4$ | $50.1 \pm 1.7$ | $53.8 \pm 0.7$ |
| | SIDCLIP$_{B/32}$ | $\mathbf{56.6} \pm 0.6$ | $\mathbf{59.2} \pm 1.4$ | $\mathbf{63.6} \pm 1.2$ | $\mathbf{67.5} \pm 0.9$ |
| | SIDCLIP$_{L/14}$ | $52.4 \pm 2.6$ | $55.4 \pm 2.1$ | $61.0 \pm 2.3$ | $66.8 \pm 0.9$ |

# 5 Discussion

## 5.1 Ablation on Method Components

Table 4 demonstrates the additional value of each SIDCLIP component and set of components. The columns "Synthesize," "Initialize," and "Distill" indicate the presence or absence of each component in each row. We select only the Flowers dataset and CLIP ViT-L/14 teacher supervision for our set of ablations for computational reasons.

We note that each of the components alone leads to an improvement over the baseline. However, not all combinations lead to an improvement. For instance, distilling to a CLIP-initialized model (Initialize ✓and Distill ✓) often underperforms distilling to the standalone model (Distill ✓). Additionally, we note that the usage of synthetic data during training (Synthesize ✓) consistently underperforms the baseline. We believe that this pattern stems from how synthetic data interacts with a small model's limited capacity. Initialization and distillation provide additional structure to bridge the teacher-student capacity gap, but synthetic data alone provides supervision the small model struggles to use effectively without that structure. This explains why synthesis is most valuable when paired with the other two components, and why the diminished contribution of synthesis in the full-shot setting is expected: when abundant real data is available, the marginal information from synthetic samples is small.

Across shots and student models, we note that all three elements of SIDCLIP are necessary to consistently obtain the best performance.

## 5.2 Additional Distillation Methods

We use KL distillation for our experiments due to its simplicity and strong performance. However, we expect SIDCLIP to be beneficial when used with a range of other distillation methods. In Table 5 we show a limited set of results indicating that SIDCLIP works synergistically with other distillation methods to obtain strong results. We show a small set of results on the Flowers dataset, using an EfficientNet B0 student and a CLIP ViT-L/14 teacher for distillation supervision. We compare the baseline, or standard finetuning, with three SIDCLIP variants: KL distillation, DKD distillation, and intermediate distillation. KL distillation is the setting we used throughout the rest of the paper. Decoupled Knowledge Distillation (DKD) (Zhao et al., 2022) is a recent SOTA distillation method which separates the standard distillation loss into two terms: a target class term and a non-target class term. We use a simple version of intermediate distillation, similar to what is proposed in (Wu et al., 2021). We generate pseudopredictions based on the features produced by the last layers of the teacher and student networks and use a KL loss between the pseudopredictions during distillation. We show that SIDCLIP leads to improved model performance, relative to the baseline, across the three distillation methods.

## 5.3 Ablation on Number of Synthetic Samples

For our main results, we use 300 synthetic samples per class for all few-shot settings, and 100 for zero-shot. In Table 7, we show a limited set of ablation experiments on the number of synthetic samples. Performance increases with more synthetic samples, but as few as 50 synthetic samples per class still yields strong results, suggesting that practitioners with tighter generation budgets could use fewer samples with modest performance loss.

## 5.4 Comparison across EfficientNet Models

Table 6 shows the Pearson and Spearman correlations between each pair of EfficientNet models. Although the EfficientNet models were trained on differing amounts of DataComp data, this comparison allows us to see whether the same patterns of correlations hold across model scales. We observe that trends are consistent across the three EfficientNet sizes and correlations between the student architectures are extremely high. Each correlation is computed across (dataset, method, shot) conditions.

Table 4: Ablations of every subset of SIDCLIP components on the Flowers dataset. Distillation supervision is from the CLIP ViT-L/14 teacher. All three components are required to obtain the best performance. Results on one random seed.

| Model | Synthesize | Initialize | Distill | 0 | 1 | 2 | 4 | 8 | Full |
|---|---|---|---|---|---|---|---|---|---|
| | | | | \multicolumn{6}{c}{**Shot**} | | | | | |
| CLIP ViT-L/14 | | | | 76.48 | 90.3 | 94.9 | 97.5 | 98.5 | 98.6 |
| EfficientNet B0 | ✗ | ✗ | ✗ | - | 34.0 | 49.8 | 68.8 | 83.5 | 86.9 |
| | ✓ | ✗ | ✗ | 1.14 | 8.4 | 11.2 | 12.3 | 14.9 | 70.7 |
| | ✗ | ✓ | ✗ | - | 50.1 | 64.0 | 78.8 | 89.7 | 91.4 |
| | ✗ | ✗ | ✓ | - | 55.4 | 71.6 | 86.6 | 91.9 | 93.1 |
| | ✓ | ✓ | ✗ | - | 11.8 | 14.4 | 14.8 | 16.1 | 85.2 |
| | ✓ | ✗ | ✓ | 3.0 | 57.3 | 64.5 | 70.1 | 78.4 | 82.1 |
| | ✗ | ✓ | ✓ | - | 65.4 | 77.0 | 88.2 | 93.3 | 94.3 |
| | ✓ | ✓ | ✓ | **10.1** | **88.5** | **88.7** | **92.6** | **94.5** | **95.7** |
| EfficientNet B1 | ✗ | ✗ | ✗ | - | 39.4 | 54.7 | 71.3 | 85.7 | 88.1 |
| | ✓ | ✗ | ✗ | 1.3 | 8.6 | 10.7 | 12.6 | 15.3 | 73.0 |
| | ✗ | ✓ | ✗ | - | 31.5 | 42.4 | 56.6 | 77.0 | 82.4 |
| | ✗ | ✗ | ✓ | - | 57.5 | 72.7 | 88.0 | 92.9 | 93.9 |
| | ✓ | ✓ | ✗ | - | 10.6 | 12.4 | 13.3 | 15.1 | 68.4 |
| | ✓ | ✗ | ✓ | 2.1 | 54.3 | 65.5 | 75.1 | 78.5 | 80.7 |
| | ✗ | ✓ | ✓ | - | 45.8 | 58.8 | 78.1 | 88.8 | 90.7 |
| | ✓ | ✓ | ✓ | **4.8** | **84.5** | **86.8** | **90.0** | **93.1** | **93.1** |
| EfficientNet B2 | ✗ | ✗ | ✗ | - | 32.3 | 50.9 | 69.7 | 84.3 | 87.7 |
| | ✓ | ✗ | ✗ | 1.3 | 8.9 | 10.5 | 13.1 | 15.1 | 69.8 |
| | ✗ | ✓ | ✗ | - | 31.8 | 42.4 | 55.2 | 75.0 | 80.1 |
| | ✗ | ✗ | ✓ | - | 55.4 | 70.4 | 86.4 | 92.4 | 93.6 |
| | ✓ | ✓ | ✗ | - | 10.4 | 14.4 | 13.6 | 15.1 | 69.2 |
| | ✓ | ✗ | ✓ | 2.6 | 57.4 | 58.7 | 73.7 | 80.2 | 80.6 |
| | ✗ | ✓ | ✓ | - | 44.5 | 45.6 | 61.0 | 88.5 | 90.9 |
| | ✓ | ✓ | ✓ | **6.5** | **85.4** | **87.4** | **91.3** | **93.9** | **94.1** |

Table 5: SIDCLIP works with several distillation methods. Results on one random seed.

| Method | 2 | 4 | 8 |
|---|---|---|---|
| | \multicolumn{3}{c}{**Shot**} | | |
| Baseline | 49.8 | 68.8 | 83.5 |
| KL Distillation | 88.7 | 92.6 | 94.5 |
| DKD Distillation | 90.0 | 92.8 | 94.2 |
| Intermediate Distillation | 87.2 | 91.5 | 93.6 |

Table 6: Correlation between EfficientNet backbones. All correlations are significant at $p < 10^{-41}$.

| | B0 vs B1 | B0 vs B2 | B1 vs B2 |
|---|---|---|---|
| Pearson | 0.994 | 0.992 | 0.997 |
| Spearman | 0.983 | 0.979 | 0.995 |

## 5.5 Computational Overhead

Synthetic data generation is expensive, but can be worthwhile in the creation of a smaller, more efficient, downstream model. We provide wall clock time references for the data generation process and for distillation using the additional synthetic images. Using one Nvidia A6000 GPU, the generation of 100 images takes 840.6 seconds (approximately 8 seconds per image). The difference in time for one epoch when using only real data as opposed to real and synthetic data can primarily be ascribed to the increased amount of data

Table 7: DTD accuracy vs. synthetic samples per class. All results are mean $\pm$ std over 3 seeds.

| Shot | Model | Synthetic Samples per Class | | | |
|------|-------|------|------|------|------|
| | | **50** | **100** | **200** | **300** |
| 8 | ViT-L/14 | $64.2 \pm 1.2$ | $65.0 \pm 0.6$ | $65.8 \pm 0.4$ | $66.8 \pm 0.8$ |
| | ViT-B/32 | $65.3 \pm 0.6$ | $65.7 \pm 0.6$ | $66.4 \pm 1.2$ | $67.5 \pm 0.7$ |
| 4 | ViT-L/14 | $59.3 \pm 2.3$ | $60.6 \pm 1.6$ | $61.3 \pm 2.0$ | $61.0 \pm 1.9$ |
| | ViT-B/32 | $62.0 \pm 0.7$ | $62.8 \pm 0.9$ | $63.4 \pm 1.7$ | $63.6 \pm 1.0$ |

being processed. For the 8 real shot setting (8 images per class), our CLIP-initialized EfficientNet B0 model takes around 24 seconds for one epoch of distillation, and the 8 real + 300 syn shot setting (308 total images per class) takes around 1021 seconds for one epoch of distillation.

### 5.6  Limitations

This work is intended as a practical guide, but faces some limitations in the scope of evaluations. Due to the setting we operate in, we only evaluate on four fine-grained image classification datasets and do not evaluate coarse-grained tasks or benchmarks that specifically evaluate distribution shift or robustness. Our student models are limited to the EfficientNet family. Despite robust results across these students, distillation to other student architectures is left to future work. This pipeline is intended to work with fixed-class classification, not general retrieval settings, which could be explored in future work. Finally, we reiterate that our method consists of a pipeline of preexisting methods so the algorithmic novelty is limited.

## 6  Conclusion

We present the SIDCLIP (Synthesize-Initialize-Distill CLIP) method, which consists of 1) augmenting the limited training data with task-specific *synthetic data* generated by using linear combinations of the CLIP image and text embeddings of existing real data; 2) *initializing* the small model as a CLIP-style model; and 3) using *knowledge distillation* to transfer more fine-grained classification information from a powerful teacher. SIDCLIP achieves the best few-shot performance on several task-specific datasets relative to existing methods, improving model performance by up to 69%, approaching and even occasionally exceeding the teacher's performance. These results hold across four datasets, two teacher architectures, three student architectures, and several few-shot settings. Our method achieves this performance by efficiently utilizing existing data to extract task-specific information from a large scale VLM such as CLIP in a resource constrained setting.

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

## A   Appendix

### A.1   Qualitative analysis of synthetic images

Figure 3 shows examples of synthetic data used in the SIDCLIP pipeline. When conditioned on one or two real images as shown in the last two columns, we can see that the synthetic images directly mirror features in the real images more than when generation is only conditioned on the caption. For instance, note the colors of the Volkswagen Beetle and the butter on the waffles.

We also note in particular that the Flowers dataset tends to yield relatively poor zero-shot performance. We can observe how much the caption-only "red ginger" image differs from both the real images and the real-image-conditioned synthetic images. Additionally, the caption-only "yellow iris" includes less background foliage. This dataset-specific discrepancy may be a contributor to the impacted zero-shot performance.

| Caption | Real Image 1 | Real Image 2 | Synthetic image generation conditioned on: | | |
| --- | --- | --- | --- | --- | --- |
| | | | caption only | caption + real image 1 | caption + both real images |
| 'Rolls-Royce Phantom Sedan 2012' | | | | | |
| 'Volkswagen Beetle Hatchback 2012' | | | | | |
| 'red ginger' | | | | | |
| 'yellow iris' | | | | | |
| 'waffles' | | | | | |
| 'spaghetti bolognese' | | | | | |

Figure 3: Synthetic images mirror the real images more closely when conditioned on real images and captions, rather than captions only.

### A.2   Details of synthetic image generation

The caption we used to produce the text embedding is always the classname. For zero shot, we use only the caption to prompt the diffusion model and provide no real image samples. For 1 shot, we use the single image in each class as the only real image sample. For 2, 4, and 8 shot, we sample two images from each class of our few shot dataset. For the full shot setting, we reuse the synthetic samples generated from the 8 shot setting.

In the 1 shot case, we use weights of 0.4 for text and 0.6 for image, and for larger shots, we use weights of 0.2 for the text and 0.4 for each image. These values were chosen via preliminary experiments.

### A.3   Hyperparameter Details

We include full hyperparameter details in Table 8.

Table 8: SIDCLIP training hyperparameters.

| Hyperparameter | Value |
|---|---|
| Optimizer | RMSProp |
| Learning rate | $8 \times 10^{-6}$ |
| Weight decay | $2 \times 10^{-4}$ |
| Batch size | 64 |
| Total epochs | 40 |
| Training schedule | 30 epochs on real + synthetic, then 10 epochs on real only |
| Distillation $\alpha$ | 0.9 |
| Distillation temperature $T$ | 2 |
| Data augmentation | RandAugment, 6 augmentations per image |
| Synthetic samples / class (few-shot) | 300 |
| Synthetic samples / class (zero-shot) | 100 |
| Embedding weights (1-shot) | $w_{\text{text}}=0.4,\ w_{\text{img}}=0.6$ |
| Embedding weights (2/4/8-shot) | $w_{\text{text}}=0.2,\ w_{\text{img1}}=0.4,\ w_{\text{img2}}=0.4$ |

