# OpenReview forum: "Leveraging Vision-Language Models for Resource Constrained Settings"
_TMLR — Accepted by TMLR_

### Review · Reviewer_UNbd · 2026-02-24

**Summary Of Contributions:**

Summary:

This paper introduces SIDCLIP, a framework for distilling a large CLIP model into a smaller one for a specific task using only a few image-class pairs (see Figure 2). First, the authors augment the limited training data by generating synthetic images via the Kandinsky generative model [1] (Section 3.1), which supports seeding generation with both text and image embeddings. A smaller student model (initialized from CLIP weights pretrained on DataComp; Section 3.2) is then trained to match the larger CLIP teacher using standard knowledge distillation (Section 3.3). In the experiments, it is shown that SIDCLIP yields well-performing strong models.

Strengths:
* S1: The method is intuitive and well-motivated. The design choices are simple yet effective, as validated by the ablation study in Table 3.

* S2: SIDCLIP yields significant performance gains for small models in the few-shot regime, demonstrating clear practical value.

* S3: The paper addresses an important real-world problem: enabling fast inference on edge devices when only minimal labeled data is available.


Weaknesses:
* W1: Each individual component (synthetic data augmentation, knowledge distillation, CLIP initialization) relies on existing techniques. While the combination is sensible, the limited conceptual novelty limits the contribution.

* W2: Experiments are conducted on only 4 datasets. Evaluation on additional benchmarks (particularly those testing distribution shift or robustness) would substantially strengthen the paper.

* W3: Section 3.3 mentions that the CLIP teacher is finetuned before distillation, but this step is not adequately discussed. On what data is the teacher finetuned? If the same few-shot examples are used, this introduces an uncontrolled confounder for attribution of the performance gains.

* W4: All experiments use EfficientNet-based student architectures. Including at least one ViT-based student would better demonstrate the generality of SIDCLIP.

* W5: In Table 1, ViT-B/32 as teacher curiously outperforms ViT-L/14 in several settings. This counterintuitive result is left unexplained.

* W6: Key hyperparameters are neither reported nor discussed. In particular, how is the weighting term $w_i​$ set (see Q2)?

* W7: Code is not provided, hindering reproducibility

---

[1] https://arxiv.org/abs/2310.03502

**Additional Comments:**

Comments:
* C1: Equations should be numbered for easier referencing.
* C2: The knowledge distillation loss appears to be missing temperature scaling inside the softmax functions. Without it, the $T^2$ factor overcompensates the gradient scaling. Since no code is provided, I could not verify whether this is only a notational oversight or an implementation bug.

Questions:
* Q1: Why do the authors compute a weighted average over the image embeddings rather than sampling directly from the n class-image pairs? That is, why not use $w_0⋅text(l_c)+w_i⋅img(x_i)$ instead of the formulation at the bottom of page 4?
* Q2: How are the values $w_i​$ chosen in the equation at the bottom of page 4?
* Q3: On what data is the teacher's linear layer finetuned? Why is finetuning necessary at all, given that CLIP is designed to work zero-shot?

**Audience:**

Yes

**Audience Explanation:**

This is a relevant, practical topic that’s of interest to some individuals in the TMLR’s audience.

**Claims And Evidence:**

Yes

**Claims Explanation:**

* The central claim (SIDCLIP produces strong small, task-specific models from only a few labeled examples) is convincingly supported by Figure 1 and Tables 1 and 2, which show clear performance gains over training the small model from scratch.

* The ablation study in Table 3 effectively isolates the contribution of each component (synthetic data generation, student initialization, and knowledge distillation).

* However, the evidence has some gaps: the evaluation spans only 4 datasets and a single student architecture (EfficientNet).

* The undiscussed teacher finetuning step and missing hyperparameter details (e.g., $w_i$) leave open questions about what is truly driving performance.

**Requested Changes:**

* Additional evaluations (W2) and non-EfficientNet student architecture, e.g., a ViT-based model (W4).

* Clarifications what data is used to finetune the CLIP teacher and its relation to the results (W3).

* Additional details on core hyperparameters, particularly $w_i$​ (W6).

* Adressing the comments and questions below.

* Minor: further investigation of W5.

---

> ### Author Response · Authors · 2026-04-25
>
> We thank the reviewer for their detailed and positive comments. We will release code with the final version of the paper to facilitate reproducibility and adoption. We address the main concerns below.
>
> 1. Additional evaluations (W2 and W4). Due to computational constraints during the rebuttal period, we were unable to pretrain a ViT-based CLIP-style student model in time, but we agree this could be interesting future work. As well, in place of additional datasets, we obtained variance results across multiple few-shot splits, which can help address the question of robustness: when we sample a different set of few-shot real data, generate synthetic data grounded in those samples, and perform distillation with the new data, the trends originally reported in Table 1 continue to hold with low standard deviations.
>
> **DTD**
>
> | Method | 1-shot | 2-shot | 4-shot | 8-shot |
> |--|--|--|--|--|
> | SIDCLIP_B/32 | 56.58 ± 0.61 | 59.15 ± 1.36 | 63.55 ± 1.22 | 67.54 ± 0.91 |
> | SIDCLIP_L/14 | 52.39 ± 2.58 | 55.43 ± 2.05 | 60.97 ± 2.29 | 66.81 ± 0.92 |
>
> **Food101**
>
> | Method | 1-shot | 2-shot | 4-shot | 8-shot |
> |--|--|--|--|--|
> | SIDCLIP_B/32 | 70.03 ± 0.45 | 72.43 ± 0.08 | 75.11 ± 0.29 | 76.95 ± 0.01 |
> | SIDCLIP_L/14 | 58.66 ± 1.54 | 64.00 ± 1.22 | 69.27 ± 0.65 | 72.78 ± 0.65 |
>
> **Flowers**
>
> | Method | 1-shot | 2-shot | 4-shot | 8-shot |
> |--|--|--|--|--|
> | SIDCLIP_B/32 | 88.00 ± 0.47 | 88.69 ± 0.06 | 89.97 ± 0.76 | 90.96 ± 0.23 |
> | SIDCLIP_L/14 | 87.64 ± 0.51 | 89.91 ± 0.40 | 93.27 ± 0.39 | 94.91 ± 0.40 |
>
>
> 2. Teacher finetuning (Q3). The teacher's appended linear classification layer is finetuned on the full training set of each downstream dataset, not on the few-shot subset. We made this choice for two reasons: it provides a high-quality teacher that does not vary across few-shot splits, isolating the effect of the SIDCLIP pipeline on the student, and it reflects a realistic deployment scenario in which a practitioner has access to a one-time, well-resourced teacher training pass before performing distillation to a deployable student. The teacher is finetuned only once per dataset and reused across all shot counts and student models. This setup also addresses a potential confounder the reviewer raises: if the teacher were finetuned on the same few-shot samples as the student, performance gains could be attributed to teacher overfitting rather than to the SIDCLIP pipeline itself.
>
> 3. Hyperparameters (Q2). We will add the following table to the appendix. The embedding weights were chosen via preliminary experiments to determine what worked well in practice. We did not observe strong sensitivity to these choices in pilot runs.
> 	| Hyperparameter | Value |
> |---|---|
> | Optimizer | RMSProp |
> | Learning rate | 8e-6 |
> | Weight decay | 2e-4 |
> | Batch size | 64 |
> | Total epochs | 40 |
> | Training schedule | 30 epochs on real + synthetic data, then 10 epochs on real data only |
> | Distillation α | 0.9 |
> | Distillation temperature T | 2 |
> | Data augmentation | RandAugment, 6 augmentations per image |
> | Synthetic samples per class (few-shot) | 300 |
> | Synthetic samples per class (zero-shot) | 100 |
> | Embedding weights (1-shot) | w_text=0.4, w_img=0.6 |
> | Embedding weights (2/4/8-shot) | w_text=0.2, w_img1=0.4, w_img2=0.4 |
>
>
> 4. Q1: We do sample directly from the available real images, and we will clarify this in Section 3.1. Specifically: for k = 1, we use the single available image for all generations; for k = 2, we use both images; for k = {4, 8}, we randomly sample 2 images per generation; for zero-shot, we use only the text embedding; and for the full-shot setting, we reuse the synthetic samples generated for the 8-shot setting. The weighted average formulation accommodates this variable number of conditioning images naturally.
>
> 5. W5: Our variance experiments confirm that ViT-B/32 as teacher often does outperform ViT-L/14, particularly at lower shot counts. We hypothesize this is due to the teacher-student capacity gap: the gap between a 307M parameter teacher and a 5.3M parameter student is much larger than between an 86M teacher and the same student, and prior work on distillation has shown that larger capacity gaps make knowledge transfer more difficult. The student may struggle to absorb the richer representations of the larger teacher, while the smaller teacher provides supervision that is closer to what the student can express.
>
> 6. Comments: We will number all equations in the final version. The reviewer is correct that our notation in Section 3.3 omits the temperature scaling inside the softmax. This is a notational oversight; the implementation applies temperature scaling within the softmax in the standard way (Hinton et al., 2015). We will correct the equation in the revision.

---

### Review · Reviewer_ffMv · 2026-03-15

**Summary Of Contributions:**

The paper introduces SIDCLIP (Synthesize-Initialize-Distill CLIP), a three-part pipeline for transferring CLIP's few-shot image classification capabilities into small, efficient models like EfficientNet Bx. The three components are:

1. generating synthetic training data using the Kandinsky diffusion model conditioned on weighted combinations of CLIP text and image embeddings;
2. initializing the student as the image encoder of a small CLIP model pretrained on DataComp;
3. performing knowledge distillation from a large CLIP teacher (ViT-L/14 or ViT-B/32) to the student.

The method is evaluated on four fine-grained classification benchmarks (StanfordCars, OxfordFlowers, Food101, DTD) in 0/1/2/4/8-shot settings, showing large improvements over standard training of EfficientNet models and outperforming TinyCLIP and TinyViT in few-shot regimes.

**Key strengths:**

* The problem setting is practical and well-motivated: deploying strong classifiers on edge devices with minimal labeled data is a real need.
* The thorough ablation study (Table 3) shows that all three components are needed in concert.
* Results are reported across multiple student architectures, teacher models, datasets, and shot counts, giving a reasonably comprehensive picture.
* The paper is clearly written and easy to follow.

**Key weaknesses:**

* The experimental evaluation is narrow in scope (only EfficientNet students).
* Key experimental details are underspecified, and reproducibility is a concern.

**Additional Comments:**

The paper addresses a sensible and practical problem, and the results are promising. The writing quality is great, and the problem is well-motivated, However, there are some gaps and questions that deserve clarification before I can confidently say “yes” that everything is evidenced. (Or claims have to be constrained accordingly.)

I would encourage the authors to address the requested changes. This paper looks like it can become a solid empirical contribution to the community.

**Audience:**

Yes

**Audience Explanation:**

The practical problem of deploying efficient classifiers with minimal labeled data by leveraging foundation models is relevant to both researchers and practitioners. The finding that combining synthetic data augmentation, CLIP-style initialization, and distillation yields large improvements over naive training is useful practical knowledge. The few-shot edge deployment scenario is increasingly common, and a well-executed practical guide would serve the community.

However, the interest is somewhat diminished by the limited scope of the evaluation. A reader might reasonably expect that combining data augmentation, good initialization, and distillation would help. The question is by how much and under what conditions, which requires more thorough experimentation than currently provided.

**Claims And Evidence:**

No

**Claims Explanation:**

More yes than no, but:

While the core claim (combining synthetic data, CLIP-style initialization, and distillation improves few-shot performance of small models) is supported directionally by the experiments, there are several issues that weaken the evidence:

1. **Missing error bars/confidence intervals.** Few-shot experiments are sensitive to which samples are selected. The paper does not report standard deviations across multiple random seeds or few-shot splits. Without this, it is impossible to assess whether the reported differences are statistically meaningful, especially in cases where margins are small (e.g., SIDCLIP$_{B/32}$ vs. SIDCLIP$_{L/14}$.

2. **Incomplete baselines.** The paper compares against standard training ("Train") and a handful of existing methods (TinyCLIP, TinyViT, Li et al. 2023), but it would be great to compare to other efficient few-shot methods such as Tip-Adapter or CLIP-Adapter (https://arxiv.org/abs/2111.03930), or CoOp (https://arxiv.org/abs/2109.01134) applied to the small CLIP-EfficientNet models, which would test whether SIDCLIP's pipeline is truly superior to parameter-efficient fine-tuning of the same small architecture.

3. **Fine-tuning the CLIP-initialized EfficientNet with real + synthetic data but no distillation** (Synthesize ✓, Initialize ✓, Distill ✗) consistently performs terribly in Table 3, which deserves more analysis. Moreover, it is somewhat understandable why Synthesize ✓, Initialize ✗, Distill ✗ underperforms, except that Synthesize ✗, Initialize ✗, Distill ✗ performs much better? How is that possible?

4. **Synthetic data generation details are sparse.** The number of synthetic samples per class is mentioned only in passing (300 synthetic in Section 4.6's timing analysis, and 100 for zero-shot in Section 4.1). How was this number chosen? Is there an ablation over the number of synthetic samples? The sensitivity of the method to this hyperparameter is unclear.

**Requested Changes:**

**Clarification:**

1. How many synthetic samples are used for 1/2/4/8 shot?
2. How many samples (real and synthetic) are used to FT the teacher? If I understand correctly for e.g. 1-shot, the teacher is also only fine-tuned with 1 sample per class + X synthetic samples that are created from the 1 sample per class + text embedding mixed randomly?

**Important:**

1. **Report variance across multiple few-shot splits.** Run at least 3 (preferably 5) random few-shot samplings and report mean ± standard deviation for the main results (Table 1) and ablations (Table 3). This is essential for interpreting the results.

2. **Correlation analysis of results.** While B0, B1, B2 cannot be directly compared with each other, a correlation analysis could be performed to validate that the experiments on these three models at least exhibit the same trends. This should be difficult to do and it would help trust the table more (it’s a lot of information to parse).

3. **Ablation over the number of synthetic samples.** How does performance change as the number of synthetic images per class varies? This is a key practical parameter that users of the method would need guidance on. For 0-shot, §4.1 states that 100 synthetic samples were used, but it is not clear how many synthetic samples are used in the other experiments.

4. **Include additional baselines.** At minimum: parameter-efficient fine-tuning methods (e.g., Tip-Adapter or CLIP-Adapter) applied to the CLIP-EfficientNet models.

5. **Clarify and unify experimental details.** Specify the exact number of synthetic samples used per class for each shot count. Report all hyperparameters (learning rate, epochs, $\alpha$, $T$ for distillation, optimizer, etc.) in a single, comprehensive table or appendix section.

**Suggested (would strengthen the paper):**

6. **Improve the framing around novelty.** The paper currently oscillates between claiming novelty ("a novel training pipeline") and positioning itself as a practical guide. Being forthright about the contribution being primarily empirical and practical would be more appropriate and would set clearer expectations.

7. **The full-shot results in Table 1 are puzzling in some cases.** For Flowers with SIDCLIP$_{L/14}$, EfficientNet B0 achieves 84.6% in the full-shot setting but 94.5% in the 8-shot setting. Is this a typo?

---

> ### Author Response · Authors · 2026-04-25
>
> We thank the reviewer for their detailed feedback and appreciation of our work.
>
> Confidence intervals: Due to compute constraints, we report variance experiments for SIDCLIP with EfficientNet B0 across DTD, Flowers, and Food in the few-shot setting, with three random seeds. Full results will be in the final version. The low variance supports the reliability of Table 1.
>
> Additional baselines: We compare SIDCLIP to Tip-Adapter and Tip-Adapter-F applied to our CLIP-EfficientNet B0 model. We report mean and standard deviation across three random seeds. SIDCLIP generally outperforms both Tip Adapter variants across datasets and number of shots.
>
> **DTD**
>
> | Method | 1-shot | 2-shot | 4-shot | 8-shot |
> |--|--|--|--|--|
> | Tip | 35.80 ± 0.57 | 40.49 ± 0.67 | 46.53 ± 1.13 | 53.23 ± 0.88 |
> | Tip-F | 36.66 ± 0.85 | 45.25 ± 0.44 | 50.06 ± 1.68 | 53.80 ± 0.65 |
> | SIDCLIP_B/32 | 56.58 ± 0.61 | 59.15 ± 1.36 | 63.55 ± 1.22 | 67.54 ± 0.91 |
> | SIDCLIP_L/14 | 52.39 ± 2.58 | 55.43 ± 2.05 | 60.97 ± 2.29 | 66.81 ± 0.92 |
>
> **Food101**
>
> | Method | 1-shot | 2-shot | 4-shot | 8-shot |
> |--|--|--|--|--|
> | Tip | 65.06 ± 0.08 | 65.14 ± 0.31 | 65.46 ± 0.11 | 65.78 ± 0.07 |
> | Tip-F | 65.19 ± 0.03 | 65.49 ± 0.35 | 66.42 ± 0.12 | 66.97 ± 0.09 |
> | SIDCLIP_B/32 | 70.03 ± 0.45 | 72.43 ± 0.08 | 75.11 ± 0.29 | 76.95 ± 0.01 |
> | SIDCLIP_L/14 | 58.66 ± 1.54 | 64.00 ± 1.22 | 69.27 ± 0.65 | 72.78 ± 0.65 |
>
> **Flowers**
>
> | Method | 1-shot | 2-shot | 4-shot | 8-shot |
> |--|--|--|--|--|
> | Tip | 75.23 ± 1.14 | 79.18 ± 1.42 | 83.65 ± 0.56 | 86.09 ± 0.37 |
> | Tip-F | 76.37 ± 1.38 | 80.24 ± 0.38 | 84.44 ± 1.28 | 91.04 ± 0.19 |
> | SIDCLIP_B/32 | 88.00 ± 0.47 | 88.69 ± 0.06 | 89.97 ± 0.76 | 90.96 ± 0.23 |
> | SIDCLIP_L/14 | 87.64 ± 0.51 | 89.91 ± 0.40 | 93.27 ± 0.39 | 94.91 ± 0.40 |
>
> **StanfordCars**
>
> | Method | 1-shot | 2-shot | 4-shot | 8-shot |
> |--|--|--|--|--|
> | Tip | 64.42 ± 0.48 | 66.41 ± 0.31 | 68.95 ± 0.54 | 72.23 ± 0.22 |
> | Tip-F | 65.39 ± 0.24 | 68.01 ± 0.41 | 70.42 ± 0.65 | 73.78 ± 0.45 |
> | SIDCLIP_B/32 | 74.6* | 76.4* | 79.4* | 81.7* |
> | SIDCLIP_L/14 | 66.4* | 73.0* | 75.9* | 80.4* |
>
> *Single seed
>
> Correlation analysis. Trends are consistent across the three EfficientNet sizes. Each correlation is computed across (dataset, method, shot) conditions. Correlations between student architectures are extremely high.
>
> | | B0 vs B1 | B0 vs B2 | B1 vs B2 |
> |--|--|--|--|
> | Pearson | 0.994 | 0.992 | 0.997 |
> | Spearman | 0.983 | 0.979 | 0.995 |
>
> All correlations are significant at p < 10^-41.
>
>
>
>
> Synthetic data ablation inconsistencies. The ambiguity stems from how synthetic data interacts with a small model's limited capacity. Initialization and distillation combine additively to bridge the teacher-student capacity gap, but synthetic data alone provides supervision the small model struggles to use effectively without that structure. This explains why synthesis is most valuable when paired with the other two components, and why the diminished contribution of synthesis in the full-shot setting is expected: when abundant real data is available, the marginal information from synthetic samples is small.
>
>
> Synthetic data details. We use 300 synthetic samples per class for all few-shot settings, and 100 for zero-shot. The DTD ablation below shows performance increases with more synthetic samples, suggesting practitioners with tighter generation budgets could use fewer samples with modest performance loss.
>
> **DTD**
>
> **8-shot**
>
> | Model | 50 | 100 | 200 | 300 |
> |--|--|--|--|--|
> | ViT-L/14 | 64.20 ± 1.17 | 64.96 ± 0.55 | 65.76 ± 0.37 | 66.81 ± 0.75 |
> | ViT-B/32 | 65.34 ± 0.63 | 65.66 ± 0.61 | 66.37 ± 1.16 | 67.54 ± 0.74 |
>
> **4-shot**
>
> | Model | 50 | 100 | 200 | 300 |
> |--|--|--|--|--|
> | ViT-L/14 | 59.28 ± 2.32 | 60.64 ± 1.55 | 61.33 ± 2.02 | 60.97 ± 1.85 |
> | ViT-B/32 | 61.95 ± 0.66 | 62.75 ± 0.85 | 63.37 ± 1.68 | 63.55 ± 1.01 |
>
> Teacher finetuning. The teacher's appended linear classification layer is finetuned once per dataset on the full training set, not on the few-shot subset, and is reused across all shot counts and student models. This provides a high-quality teacher that does not vary across few-shot splits, isolating the effect of the SIDCLIP pipeline on the student.
>
>
> | Hyperparameter | Value |
> |--|--|
> | Optimizer | RMSProp |
> | LR | 8e-6 |
> | WD | 2e-4 |
> | BS | 64 |
> | Epochs | 40 |
> | Training schedule | 30 epochs on real + synthetic data, then 10 epochs on real data only |
> | Distillation α | 0.9 |
> | Distillation temp. T | 2 |
> | Embedding weights (1-shot) | w_text=0.4, w_img=0.6 |
> | Embedding weights (2/4/8-shot) | w_text=0.2, w_img1=0.4, w_img2=0.4 |
>
> Novelry. We will revise claims about novelty in the paper and include additional details, including releasing code, so as to improve the practical usage of our work.
>
> Anomalous Result. This result was a typo. The correct value is 95.71%.

---

### Review · Reviewer_W2k5 · 2026-04-11

**Summary Of Contributions:**

The paper proposes a three-stage pipeline, SIDCLIP, for developing VLM-based classifiers in resource-constrained settings: synthetic data generation, small CLIP initialization, and knowledge distillation. Through extensive experiments across diverse classification benchmarks and multiple model architectures, the paper demonstrates consistent performance improvements. However, as the authors themselves acknowledge, each of these components has already been explored in prior work, and the paper does not appear to introduce a fundamentally new algorithmic component or a particularly novel combination of ideas. Moreover,  I am also not fully convinced about the significance of the problem setting itself (synthesizing images and then training a smaller model for a few-shot classification scenario).

**Audience:**

No

**Audience Explanation:**

My main concern is the paper’s overall contribution. I do not find a clearly unique combination of approaches here. Since the largest performance gains seem to come primarily from knowledge distillation, while the other two components appear closer to straightforward implementation choices or applications of existing ideas, I am not fully convinced that the paper delivers a sufficiently new contribution to the community.
In other words, although the reported empirical results are impressive and the method may be practically useful, the paper does not seem to offer enough novelty or deeper technical insight to constitute a strong standalone research contribution.
I also find the practical significance of the problem setting somewhat unclear. The paper focuses on generating synthetic images to train a smaller model in a few-shot classification regime, but it is not fully convincing why this particular setup is sufficiently important or broadly relevant in practice, especially given the resource-constrained setting and the goal of deploying a small model.

**Broader Impact Concerns:**

Wrote above.

**Claims And Evidence:**

Yes

**Claims Explanation:**

The paper shows consistent improvements across diverse settings, including multiple student models, teacher models, and classification benchmarks, which supports the effectiveness of the method. The ablation studies are thorough, and the analyses of alternative distillation methods and computational overhead are also valuable.

That said, it remains unclear whether the proposed pipeline can extend beyond classification to more general retrieval settings. In addition, the role of the synthesis component is somewhat ambiguous, as it appears to be harmful in some cases relative to the base model, and its contribution in the full-shot setting seems limited compared to the other components. More explanation on these points would strengthen the paper.

**Requested Changes:**

Wrote above.

---

> ### Author Response · Authors · 2026-04-25
>
> We thank the reviewer for their thoughtful comments and are glad they found the claims in the paper to be generally well supported. We agree the novelty lies in the combination of pre-existing components and the demonstration of this particular pipeline's efficacy, rather than in a fundamentally new algorithm. We address three of the reviewer's specific concerns below.
>
> 1. " [I]t remains unclear whether the proposed pipeline can extend beyond classification to more general retrieval settings". By design, SIDCLIP produces compact, task-specific classification models, and is targeted at users who know their downstream task and want to trade general-purpose capability for efficiency. Users who need to preserve general retrieval capability may be better served by methods like TinyCLIP or MobileCLIP, which are designed for exactly that goal. Extending SIDCLIP to retrieval would require some modifications, notably since the synthesis and distillation steps are tied to particular classes, but could be an interesting direction for future work.
>
> 2. "[T]he role of the synthesis component is somewhat ambiguous, as it appears to be harmful in some cases relative to the base model, and its contribution in the full-shot setting seems limited compared to the other components." The ambiguity the reviewer notes in the synthesis results stems from how synthetic data interacts with a small model's limited capacity. Unlike initialization and distillation, which combine roughly additively to bridge the teacher-student capacity gap, synthetic data alone provides supervision that the small model struggles to use effectively. Without the structure imposed by a CLIP-style initialization and the targeted signal from a distillation loss, training a small model on large quantities of synthetic data appears to degrade rather than improve performance. This explains why synthesis is most valuable when paired with the other two components, and why the full SIDCLIP pipeline outperforms any subset. The diminished contribution of synthesis in the full-shot setting is consistent with this picture: when abundant real training data is available, the marginal information added by synthetic samples is small, and the pipeline's gains come primarily from initialization and distillation. Synthesis is most useful precisely in the few-shot regime our method targets.
>
> 3. "It is not fully convincing why this particular setup is sufficiently important or broadly relevant in practice." The setting we target, with few labeled examples, a tight inference budget, and a fixed downstream task, arises naturally when data collection is expensive and edge inference is required. Some examples of this include consumer mobile applications and smart home devices, where practitioners cannot collect ImageNet-scale labeled data for their specific task, cannot run a 300M parameter model at inference time, and do not need general-purpose recognition. Existing approaches force a trade-off: deploy a large CLIP model (too expensive), distill a general-purpose small model and accept lower task-specific accuracy (TinyCLIP, MobileCLIP), or train a small model from scratch (which our results show performs poorly). SIDCLIP is designed for exactly this gap. We agree the contribution is practical rather than algorithmically novel, and we believe the consistent gains across four datasets, three student architectures, and multiple teachers demonstrate meaningful value for practitioners working in this regime.

---

### Decision · Action_Editor_JVUi · 2026-05-19

**Recommendation:** Accept with minor revision

**Additional Comments:**

The responses promised some changes, e.g., revising novelty claims (ffMv, W2k5), additional results (UNbd), variance on the results (ffMv), and additional details (ffMv, UNbd, W2k5). These changes are detailed in the response but have not yet been implemented with a revision of the manuscript: before the final acceptance, the feedback should be integrated together with the promised updates.

**Audience:**

Yes

**Audience Explanation:**

The article proposes an efficient approach for image classification with vision-language models. Given the widespread use of these models and the need to reduce their computational constraints, this article may be of interest to researchers and practitioners in computer vision and multimodal learning. While the technical contribution is limited, the role of synthetic data and its combination with the distillation objective (Tab. 3) provides interesting insights for future work.

**Claims And Evidence:**

Yes

**Claims Explanation:**

The core claim of the article is that the proposed approach can improve the performance of small, efficient models on zero-/few-shot classification tasks. This has been mostly verified experimentally (e.g., Tab. 1-2) and ablation studies (e.g., Tab. 3). The initial concerns on the reviewers (e.g., baselines, variance, single backbone) were addressed in the rebuttal, expanding the experimental validation.